# Comparison of Gas Treatments of High Oxygen, Carbon Monoxide, and Nitric Oxide on Ground Beef Color in Modified Atmosphere Packaging

**DOI:** 10.3390/foods13060902

**Published:** 2024-03-16

**Authors:** Benjamin J. Carpenter, Thomas W. Dobbins, Manuel Sebastian Hernandez, Samantha N. Barker, Kaitlyn R. Loomas, Wesley N. Osburn, Jerrad F. Legako

**Affiliations:** 1Department of Animal and Food Sciences, Texas Tech University, Lubbock, TX 79409, USA; benjamin.carpenter@ttu.edu (B.J.C.); tdobbins@ttu.edu (T.W.D.); sebastian.hernandez@ttu.edu (M.S.H.); sam.barker@ttu.edu (S.N.B.); kloomas@ttu.edu (K.R.L.); 2Department of Animal Science, Texas A&M University, College Station, TX 77843, USA; wesley.osburn@ag.tamu.edu

**Keywords:** carbon monoxide, ground beef, meat color, modified-atmosphere packaging, nitric oxide

## Abstract

The objective of this study was to evaluate the viability and performance of nitric oxide modified-atmosphere packaging (MAP) as a novel alternative to high oxygen and carbon monoxide MAP for ground beef. Packages of ground beef under high oxygen (HI-OX), carbon monoxide (CO), and nitric oxide (NO) atmospheres were evaluated for descriptive and instrumental color every 12 h during a 120 h display period. Surface myoglobin percentages, internal cooked color, thiobarbituric acid reactive substances (TBARS), and residual nitrite and nitrate were also evaluated. There were gas × time interactions for descriptive color, discoloration, a* values, b* values, deoxymyoglobin percentages, and metmyoglobin percentages (*p* < 0.05). There were also gas-type main effects for cooked color and TBARS (*p* < 0.05). Carbon monoxide maintained the most redness and least discoloration throughout the display period, while HI-OX started with a bright red color but rapidly browned (*p* < 0.05). Nitric oxide started as dark red to tannish-red but transitioned to a dull red (*p* < 0.05). However, NO had increased redness and a* values for internal cooked color (*p* < 0.05). Although CO outperformed NO packages, NO exhibited a unique color cycle warranting further research to optimize its use.

## 1. Introduction

Meat color remains the primary quality attribute for a consumer’s willingness to purchase fresh retail meat products. A consumer perceives a meat product (i.e., beef) as “fresh” if it is bright cherry red; therefore, a product’s perceived freshness can be maintained by reducing oxidative browning [1]. Although many packaging technologies and pre-treatments exist to increase shelf life and reduce oxidative browning, modified-atmosphere packaging (MAP) has seen consistent application for fresh meat products over the last few decades [2]. The two gas compositions that have seen the most usage within MAP are high-oxygen (HI-OX) and carbon monoxide (CO) [1].

High-oxygen MAP commonly uses a gas composition of 60–80% oxygen with 20–30% carbon dioxide, which promotes oxymyoglobin formation to develop a desirable bright cherry-red color. Furthermore, the concentration of carbon dioxide provides an antimicrobial effect against the growth of Gram-negative spoilage bacteria, thus increasing the product’s shelf life [1]. However, there are multiple concerns with the use of HI-OX packaging. Lipid and protein oxidation can be accelerated by a higher oxygen concentration, leading to the production of volatile off-flavors and aromas as well as decreased tenderness [3,4,5]. Additionally, premature browning may occur during cooking resulting in a cooked color appearance before reaching the required temperature to reduce pathogenic bacteria to safe levels [6]. An anaerobic alternative to HI-OX is CO, which typically uses a gas composition of ~0.4% carbon monoxide and 40–60% carbon dioxide, balanced with nitrogen [7]. Carbon monoxide MAP provides a bright red color with high color stability due to the formation of carboxymyoglobin [8]. Furthermore, the gas composition of CO packages creates an anaerobic environment contributing to an extended shelf life [9]. Despite this, the use of CO for MAP has been greatly debated due to its potential toxicity and ability to mask visible spoilage. Although spoilage can also be assessed through odor, this does not occur until after the consumer has purchased and opened the package. However, these issues have been addressed by studies, which found that 0.4% CO is not likely to be toxic [10,11] or mask visible spoilage [12,13]. 

Despite CO not being able to mask spoilage, it still inaccurately presents freshness and shelf life through its high color stability and redness beyond the marked shelf life, as seen in a study by Jayasingh et al. [14]. Due to consumers making more decisions concerning the health and nutrition of their food, there has been an increase in demand for fresher, less processed foods. This is evident from a study by Grebitus et al. [15], which found that although consumers in the U.S. preferred brighter red packages of ground beef, their willingness to purchase those same packages decreased once they were informed on the use of CO. Carbon monoxide and HI-OX have been the most predominately used for MAP in the beef industry; however, other food industries have seen the use and research of other novel gas types. For instance, a study from Wang et al. [16] found that tilapia filets increased in quality attributes and color stability when packaged with 0.4% nitric oxide (NO). Nitric oxide is a ligand that can bind to the sixth coordinate site of myoglobin, forming red nitrosyl myoglobin, as well as the development of the cured meat pigment, nitrosylhemochrome, upon cooking [17]. For the beef industry, NO as a MAP gas type has not been thoroughly researched, and there is disagreement if NO is able to directly bind to heme iron, or if it needs to form nitrite ions under aerobic conditions [17]. Nitric oxide could more accurately reflect the freshness of beef while improving color, and it may have more antioxidant capacity. However, NO MAP could face issues associated with nitrite-embedded packaging such as increased residual nitrite and nitrate, and an underdone cooked color known as persistent pinking due to the presence of nitrosylhemochrome [18]. Because of this, the objective of this study was to evaluate the viability and performance of NO MAP as a novel alternative to HI-OX and CO MAP for ground beef. 

## 2. Materials and Methods

### 2.1. Package Preparation and Display

Packaging treatments varied by 3 different gas compositions comprised of (1) 80% oxygen with 20% carbon dioxide (HI-OX); (2) 0.4% carbon monoxide, 30% carbon dioxide, and balanced with nitrogen gas (CO); and (3) 0.4% nitric oxide balanced with nitrogen gas (NO).

Fifteen 4.5 kg chubs of commodity 80/20 ground beef were purchased from a local purveyor. They were then reground at a 7 mm grind size and formed into 0.45 kg bricks using a Biro heavy horsepower grinder (Biro Manufacturing Company, North Canton, OH, USA) with a block attachment. The last 5 bricks of each chub were selected to prevent contamination from the prior chub (*n* = 5 bricks/chub), and each chub was randomly assigned a gas type (*n* = 5 chubs/gas type). Two of the 5 bricks per chub were randomly selected to display descriptive color and instrumental color analysis. The other 3 bricks per chub were randomly assigned a time at 0 h, 72 h, and 120 h of display to be opened, vacuum-packaged, and frozen at −20 °C until further analyses. Bricks were packaged with a Dri-Loc moisture-absorbing pad in Cryovac black plastic MAP trays with clear, multi-layer barrier film (LID 1050 film, OTR < 20 cc^3^ oxygen/m^2^/24 h at 4.4 °C and 100% relative humidity; Cryovac Sealed Air, Duncan, SC, USA) a semiautomatic sealing machine (G. Mondini CV/VG-S Semi-Automatic 320 × 500 with Vacuum and Gas, Harpak, Easton, MA, USA). The packages were equilibrated in the dark under refrigeration (2 to 4 °C) for 48 h prior to display. 

The packages were displayed in coffin-style retail cases (M1-GEA, Hussmann, Bridgeton, MO, USA) for 120 h under continuous florescent lighting with lux of the lighting measured at the surface of the products across both cases during the display period (1500 ± 223 lux). Temperature was also monitored every 12 h (1.4 ± 1.3 °C) across both cases during the display period.

### 2.2. Descriptive Color and Discoloration

Descriptive color and discoloration were evaluated every 12 h during the 120 h display period starting with 0 h. Descriptive color was evaluated by 6 trained panelists using the AMSA 8-point scale for ground beef display discoloration [17], as seen in Table 1. Panelists also evaluated discoloration on a percentage basis. Panelists were trained for objective evaluation in accordance with the 2023 AMSA color guidelines [17]. 

### 2.3. Instrumental Color and Surface Myoglobin Determination

Instrumental color was evaluated every 12 h during the 120 h display period. The L* (black to white), a* (green to red), and b* (blue to yellow) values were analyzed using a Hunterlab Miniscan EZ 4500 (Hunter Associates Laboratory, Inc. Reston, VA, USA) with a 45°/0° directional viewing geometry, 31.8 mm port, and 25 mm viewed area. Wavelength absorbances at 470, 530, 570, and 700 nm were also evaluated for conversion to percent surface deoxymyoglobin (DMb), oxymyoglobin (OMb), and metmyoglobin (MMb) using the Krzywicki method [19]. Nitrosyl myoglobin (NO-Mb) formation for NO packages was evaluated as a ratio of absorbances at 650 and 570 nm using the methodology of Smith et al. [18]. Oxymyoglobin and NO-Mb were only evaluated for HI-OX and NO packages, respectively, due to the interference of other forms of myoglobin. Three scans were taken across each sample and averaged for all measurements evaluated.

### 2.4. Internal Cooked Color

The bricks frozen from time points at 0 h, 72 h, and 120 h of retail display were thawed in refrigeration 24 h prior to cooking. Bricks were made into 150 g patties using a patty press and cooked to a target temperature of 71 °C on a flat-top gas grill. Patties were flipped approximately every minute, and temperature was evaluated using a Cooper-Atkins 351 AquTuff thermocouple (Cooper-Atkins, Middlefield, CT, USA). After reaching the target temperature, patties were removed from the grill and cut down the middle for internal cooked color evaluation. The average peak temperature of the patties was 75.1 ± 5.8 °C. 

Internal cooked color was evaluated by 6 trained panelists using the AMSA 7-point scale for internal cooked color [17], as seen in Table 2. Instrumental color data of a*, b*, and L* were also collected for instrumental cooked color using a Hunterlab Miniscan EZ 4500. Three scans were taken along the cross-sections of each sample and averaged.

After internal cooked color evaluation, both cooked samples and the remainder of raw samples for each brick were flash-frozen in liquid nitrogen and homogenized (NutriBullet LEAN, Pacoima, CA, USA) for chemical analyses. Powdered homogenates were stored in individual Whirl-Pak bags at −80 °C.

### 2.5. Thiobarbituric Acid Reactive Substances 

Thiobarbituric acid reactive substances (TBARS) values (mg malondialdehyde/kg meat homogenate) were determined by procedures stated by Buege and Aust [20] and modified by Luque et al. [21]. Briefly, 5 g of raw powdered homogenate was blended (Fisher-brand 850 homogenizer, Fisher Scientific, Pittsburgh, PA, USA, at 5000 rpm for 30 s) with 15 mL of distilled water. Samples were then centrifuged at 1850× *g* for 10 min. Then, 2 mL of supernatant from each sample was then transferred and combined with trichloroacetic acid, thiobarbituric acid, and butylated hydroxyanisole. Samples were then heated in a water bath at 100 °C for 15 min, then chilled in an ice bath for 10 min. Samples were then centrifuged again at 1850× *g* for 10 min. The supernatants were then extracted and plated on a 96-well plate with absorbances read at 531 nm. Calculations were determined by a standard curve made with 1,1,3,3-tetra-ethoxypropane.

### 2.6. Residual Nitrite and Nitrate

Powdered homogenates of raw and cooked samples were sent to the Texas A&M Department of Animal Science for residual nitrite and nitrate analysis. Ethylenediaminetetraacetic acid (EDTA) and N-ethylmaleimide (NEM) were mixed with Dulbecco’s Phosphate Buffer Saline (DPBS) to molar concentrations of 2.5 mM and 10 mM. Then, 3 g of powdered raw and cooked sample was homogenized with 27 mL of the DPBS solution using a Polytron^®^ homogenizer (Kinematic AG, Malters, Switzerland) at 10,000 rpm for 1 min. The homogenates were split into different 50 mL centrifuge tubes and centrifuged for 10 min at 10,000× *g* at 4 °C using a JA-14.50 rotor and Avanti JXN-26 centrifuge (Beckman Coulter, Maryfort, Ireland). After centrifugation, 400 µL of supernatant was transferred to 2 mL centrifuge tubes with 400 µL of methanol and vortexed for 3–5 s. After sitting at 4 °C for 10 min, tubes were centrifuged for 10 min at 13,000× *g* at 4 °C using an Eppendorf microcentrifuge (Eppendorf, Framingham, MA, USA). Aliquots of 100 µL of the supernatant were then transferred to 96-well plates for analysis using an ENO-30 nitrate/nitrite analyzer (Amuza Inc., San Diego, CA, USA) equipped with an autosampler (Eicom, Barneveld, The Netherlands). Nitrite and nitrate were measured utilizing a colorimetric diazo coupling method and HPLC compared to standard calibration curves using 0.1 to 16 ppm of sodium nitrite and sodium nitrate. The area under the curve for nitrite and nitrate was determined via Clarity Chromatography Solutions software (Version 8.6.1.69, Amuza Inc., San Diego, CA, USA),and the values were then calculated via Excel 2016 (Microsoft Corporation, Redmond, WA, USA) Samples that did not have detectable amounts of nitrite or nitrate were included in statistical analyses as the lowest ppm detected.

### 2.7. Data Analysis

Data were analyzed using the GLIMMIX procedure of SAS version 9.4 (Cary, NC, USA). Data were analyzed as randomized block design with repeated measures for gas type, display time, and their interaction as fixed effects, while chub served as the block. The Kenward–Roger adjustment was also used to estimate denominator degrees of freedom. The covariance structure showing the lowest Akaike information criterion was used. Significance was determined at alpha ≤ 0.05. Internal cooked color, TBARS, residual nitrite, and residual nitrate data were analyzed with the same procedure but without repeated measures. Raw/cook treatment and its interactions with gas type and display time were included as fixed effects for the analysis of residual nitrite and nitrate. 

## 3. Results 

### 3.1. Descriptive Color and Discoloration

There was a gas × time interaction for descriptive surface color scores (Figure 1a; *p* < 0.001) where HI-OX and CO increased in color score over the display period while NO decreased in color score (*p* < 0.05). High-oxygen packages were bright red at 0 to 24 h but began rapidly browning at 48 (*p* < 0.05). By 84 to 120 h, HI-OX packages were tan to brown (*p* < 0.05). Carbon monoxide packages maintained a bright red to dull red color for most of the display period but were slightly dark red at 108 and 120 h (*p* < 0.05). Nitric oxide packages were moderately dark red at 12 and 24 h but transitioned to a dull red by 120 h (*p* < 0.05). 

Percent discoloration had a gas × time interaction (Figure 1b; *p* < 0.001) where CO had the least discoloration throughout the display period compared to all other treatments, but still increased overall (*p* < 0.05). High-oxygen and CO were similar at 0 h (*p* > 0.05), but HI-OX rapidly increased at 60 h with peak discoloration at 96 h (*p* < 0.05). Nitric oxide had the most discoloration initially but decreased in discoloration over the display period (*p* < 0.05). 

### 3.2. Instrumental Color (L*, a*, and b*)

There was a gas × time interaction for a* values (Figure 2a; *p* < 0.001) where CO and HI-OX had greater initial a* values. However, HI-OX decreased over the display period while CO maintained greater a* values (*p* < 0.05). Nitric oxide packages started with the lowest initial a* values but increased over the display period with greater a* values than HI-OX for the 72–120 h time points (*p* < 0.05). 

There was a gas × time interaction for b* values (Figure 2b; *p* < 0.001) where HI-OX packages at 0 to 24 h were the greatest in b* values but decreased over the display period and had the least b* values at 120 h (*p* < 0.05). Carbon monoxide packages decreased in b* values over the display period, while NO packages increased in b* values (*p* < 0.05). 

There was a gas-type main effect observed for L* values (Figure 2c; *p* < 0.001). L* values were greater in CO, intermediary in HI-OX, and least in NO (*p* < 0.05). Additionally, a display time effect was observed (Figure 2d; *p* = 0.002), where the greatest L* values were observed in packages at 0 h and the least L* values at 60 and 72 h (*p* < 0.05).

### 3.3. Surface Myoglobin Percentages

There was a gas × time interaction for calculated DMb% (Figure 3a; *p* < 0.001) where CO packages had the most DMb% initially and did not change over the display period (*p* < 0.05). High-oxygen packages had greater DMb% than NO initially, but HI-OX decreased in DMb% over the display period while NO increased in DMb% (*p* < 0.05).

Because the oxymyoglobin percentage is calculated by the difference between DMb% and MMb% from total myoglobin content, and since there could be other forms of myoglobin present in CO and NO such as carboxymyoglobin and nitrosyl myoglobin, oxymyoglobin was only calculated for HI-OX. A main effect of display time occurred for the percentage of OMb in HI-OX packages (Figure 3b; *p* < 0.001). High-oxygen packages decreased in OMb% over the display period (*p* < 0.05). 

There was a gas × time interaction for MMb (Figure 3c; *p* < 0.001) where initial percentages of MMb were similar across all gas types (*p* > 0.05). Carbon monoxide had a slight increase in MMb%, but HI-OX greatly increased having the most MMb% at 120 h (*p* < 0.05). Nitric oxide decreased in DMb% over the display period (*p* < 0.05). 

Nitrosyl myoglobin formation is expressed as the ratio of absorbance at 650 nm ÷ absorbance at 570 nm. Because other forms of myoglobin such as carboxymyoglobin and oxymyoglobin can interfere with absorbances at wavelengths of 650 and 570 nm, the ratio of NO-Mb formation was only evaluated for NO packages. There was a main effect of display time for NO-Mb formation in NO packages (Figure 3d; *p* < 0.001). The ratio of NO-Mb formation increased over the display period and was the greatest at 96–120 h (*p* < 0.05).

### 3.4. Internal Cooked Color

There was a gas-type main effect for cooked color scores (Figure 4a; *p* < 0.001) where NO packages had the least cooked color score with a pink to slightly pink internal color (*p* < 0.05). Carbon monoxide and HI-OX packages were similar in score with a pinkish-gray to grayish-tan/brown internal color (*p* > 0.05). There was not a main effect of display time for internal cooked color scores (*p* = 0.365).

There was a gas-type main effect for cooked color a* values (Figure 4b; *p* < 0.001) where CO and HI-OX packages were similar (*p* > 0.05). However, NO packages had increased a* values (*p* < 0.05). Display time did not have a main effect for cooked color a* value (*p* = 0.781). There were no main effects of gas type or display time for cooked color L* values or b* values (*p* > 0.05).

### 3.5. Thiobarbituric Acid Reactive Substances

There was a gas-type main effect observed for mg malondialdehyde/kg meat homogenate (Figure 5; *p* < 0.001), but no effect was observed for display time (*p* = 0.0511). High-oxygen packages had the most malondialdehyde, CO had an intermediate amount, and NO had the least malondialdehyde (*p* < 0.05).

### 3.6. Residual Nitrite and Nitrate

There was a gas × raw/cook interaction for ppm residual nitrite (Figure 6a; *p* = 0.009) where nitrite increased in all gas types after cooking (*p* < 0.05). For cooked samples, NO had the greatest nitrite concentration while CO had the least nitrite (*p* > 0.05). Additionally, the ppm nitrite of raw NO samples was similar to the ppm nitrite of cooked CO samples (*p* > 0.05). Overall, raw HI-OX and CO samples had the least amount of nitrite (*p* < 0.05). There was also a tendency for a display time × raw/cook interaction for ppm residual nitrite (*p* = 0.084) where cooked samples at 120 h of display time had the most nitrite (*p* < 0.05).

There was a gas × display time × raw/cook interaction for ppm residual nitrate (Figure 6b; *p* < 0.001) where cooked NO samples at 120 h had the most nitrate (*p* < 0.05). Cooked NO samples also had the most nitrate among gas types and cook treatment for the 0 and 12 h time points (*p* < 0.05). While the cooked samples for all gas types increased in nitrate from 72 h to 120 h (*p* < 0.05), the raw samples for all gas types did not change over the time points (*p* > 0.05). Furthermore, the raw samples for all gas types were not significantly different from each other (*p* > 0.05). 

## 4. Discussion

Throughout the display period, CO packages outperformed NO and HI-OX packages in many regards. Carbon monoxide packages maintained the most redness and a* values while having the least discoloration compared to the other two packaging types. Although HI-OX started with similar color scores and percent discoloration as CO, these packages rapidly browned and discolored at around 48 h into the display period. The high color stability of CO packages is well understood in the literature; CO has 28–51 times the affinity to the iron-porphyrin site of myoglobin than oxygen, causing carboxymyoglobin to be more stable than oxymyoglobin [8]. This affinity makes carboxymyoglobin more resistant to oxidizing to MMb. However, the binding of CO does not permanently fix the color of ground beef. Hunt et al. [22] found that, generally, an increase in storage and display time decreased a* values in all MAP packages. Similarly, CO packages in this study had a slight decrease in a* and redness while increasing in MMb and discoloration over the display period. The initial color score and a* values in HI-OX packages were similar to CO packages due to oxidative bloom promoting the initial formation of OMb, which is similar in redness to carboxymyoglobin [7]. However, the OMb in HI-OX packages quickly deteriorates to MMb causing a rapid increase in discoloration and a decrease in redness and a* values [17]. 

Nitric oxide packages started as dark red to tannish-red with less a* values and more discoloration. However, NO transitioned to a dull red and decreased in discoloration over the display period, approaching similar values for CO and outperforming HI-OX for the second half of the display period. This indicates the presence of an extended lag phase for nitric oxide’s color cycle, which is also reflected by the surface myoglobin percentages for NO. All packaging treatments started with similar values of MMb. High-oxygen packages rapidly increased in MMb at 48 h due to a net decrease in DMb and OMb. This increase in MMb for HI-OX packages paralleled its decrease in redness and a* values. It is known that OMb is primarily responsible for the redness of beef, and that its deterioration to MMb is responsible for browning and discoloration [17]. Although NO packages had similar initial amounts of MMb compared to CO packages, CO still had more initial a* value and was a bright red while NO was dark red to tannish red. Nitric oxide as opposed to HI-OX and CO can reversibly bind with ferric heme, forming nitrosyl metmyoglobin (NO-MMb), although it has a significantly higher affinity for ferrous heme [23]. The low levels of MMb in NO packages, despite visual browning and discoloration, could be due to the presence of NO-MMb. Nitrosyl metmyoglobin would likely have a different corresponding wavelength than normal MMb; therefore, it would not be detected by the Krzywicki method [19]. The equilibrium of NO-MMb likely plays a primary role in the packages’ lag phase. Nitrosyl metmyoglobin is reduced to nitrosyl myoglobin in the presence of exogenous and endogenous reductants such as NADH [24,25]. An alternative pathway may also exist for the auto-reduction of NO-MMb into NO-Mb [26]. Initial browning of NO packages is likely due to the presence of NO-MMb, which reduces to a redder NO-Mb during the display period causing an extended lag phase. The reduction of NO-MMb to NO-Mb is supported by the increase in the ratio of NO-Mb formation over the display period. The increase in the NO-Mb ratio also parallels the increase in redness and a* values for NO packages.

Nitric oxide packages had more redness for internal cooked color than the other packaging types. The cooked color for NO packages was pink, while CO and HI-OX packages were pinkish-gray to grayish-tan/brown. This is also confirmed by instrumental color scores for internal cooked color, which demonstrated higher a* values in NO patties. This indicates an issue of persistent pinking is a problem in beef associated with high pH values; however, persistent pinking is also an issue in cured meat due to the presence of nitrosylhemochrome [17,27]. Nitrosylhemochrome is formed when NO-Mb is exposed to heat; therefore, it is likely the cause of the increased redness of the internal cooked color in NO packages because the presence of NO-Mb and/or NO-MMb is expected when myoglobin is exposed to nitric oxide [17]. Persistent pinking presents a problem for NO packages since many consumers base the degree of doneness and cooking time on cooked color [17]; therefore, future research and use of NO MAP needs to be optimized to minimize the effect of persistent pinking.

Nitric oxide packages had less mg malondialdehyde/kg meat homogenate than CO and HI-OX packages. Malondialdehyde is a product of lipid oxidation; therefore, its quantification through TBARS is used as a measurement of lipid oxidation. Nitric oxide packages likely had less malondialdehyde due to nitric oxide’s capacity as an antioxidant. However, the literature has stated that increased levels of residual nitrite and nitrate can underestimate values for TBARS, and therefore the treatment effect of NO may not be fully accurate [28,29]. 

Nitric oxide packages had significantly more residual nitrites and nitrates than CO and HI-OX for both raw and cooked samples; however, residual nitrite and nitrates were present in all gas types despite CO and HI-OX not having any added nitrogen. The nitrite and nitrate found within these packages could be due to a base level innate to beef [30]. The detection of nitrites and nitrates could also be due to possible packaging contamination or trace amounts of gases remaining in the lines of the modified-atmosphere packaging machine. The increase in nitrate over display time and from cooking can be explained through increased oxygenation from increased exposure and heat [30]. While some cooking methods such as boiling have been shown to decrease nitrate levels, a study from Iammarino et al. [30] has demonstrated that grilling can increase the amount of nitrite and nitrate. An increase in nitrite and nitrate for NO packages is also expected. The nitric oxide that binds with myoglobin can later dissociate to form nitrate, and the nitrate can reduce to nitrite [24]. Carbon monoxide binds with ferrous myoglobin perpendicular to the porphyrin plane, but NO becomes partly anionic when binding with myoglobin, causing a bent structure [24]. This structure allows the binding of oxygen to form a superoxide ferric complex. This is a non-reversible intermediate, eventually leading to the formation of metmyoglobin and nitrate [24]. Alternatively, NO bonded with ferrous heme could be reversibly replaced with oxygen [31]. The rate of this pathway would increase during cooking due to the introduction of oxygen and heat, which would explain the increase in nitrite and nitrate for cooked NO samples [24]. Residual nitrite and nitrate are a concern in meat due to their association with carcinogenic nitrosamines [32]. However, the amount of residual nitrite and nitrate in NO packages was still within the acceptable range for what is recognized as safe by the FDA with nitrite below 200 ppm and nitrate below 500 ppm [33].

## 5. Conclusions

Nitric oxide packages had less initial redness and greater initial discoloration, but redness increased and discoloration decreased over time. On the other hand, CO and HI-OX started with more initial redness and less discoloration, but both gas types decreased in redness and increased in discoloration over time. While CO and HI-OX initially outperformed NO in the display period, CO and NO had similar redness and discoloration toward the end of the display, and both gas types outperformed HI-OX, which rapidly browned and discolored after 72 h of display. Additionally, the presence of persistent pinking in NO-cooked patties and an increase in residual nitrates and nitrites causes NO to be less desirable for ground beef packaging. For this display setting and these gas formulations, CO performed better than NO and HI-OX for the majority of the display period and would be preferred to maintain redness.

Although CO would still be preferable over NO to maintain redness for MAP ground beef, NO still has viability as a MAP gas type due to its unique color cycle, warranting further research to optimize its concentration and use. Due to NO’s extended color cycle lag phase, a longer storage or transport period could cause packages to possibly transition to a more desirable visual color before display in retail cases. Different concentrations of NO gas could also improve the initial redness and discoloration since the formation of NO-myoglobin is dependent on ligand binding determined by the bimolecular rate constant and the concentration of myoglobin and NO [34]. Furthermore, lower concentrations could decrease the amount of persistent pinking and decrease the amount of residual nitrite and nitrate. Along with an antioxidant effect for NO indicated by decreased TBARS values, NO could have a possible antimicrobial effect similar to nitrites’ reduction of *Clostridium botulinum*, which should be investigated [32]. The pathway of the reduction of NO-MMb to NO-Mb also needs further investigation to better understand the mechanism for the color cycle of nitric oxide and myoglobin. 

## Figures and Tables

**Figure 1 foods-13-00902-f001:**
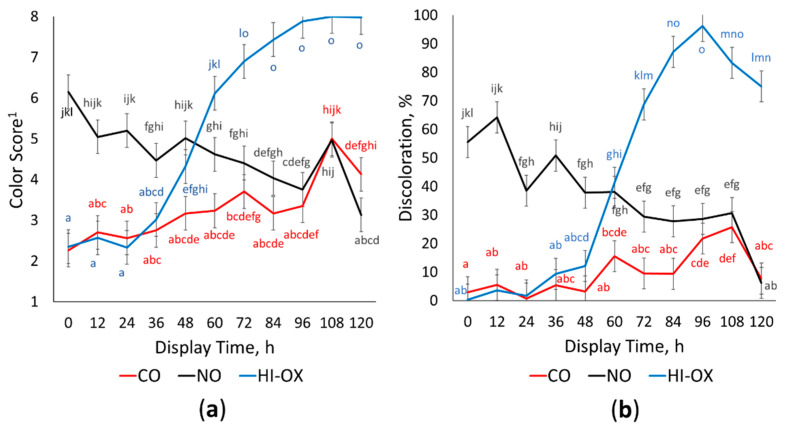
Descriptive color and discoloration. LS means with the same superscript are not significantly different (*p* > 0.05): (**a**) Least squares means of descriptive color scores for 0.4% carbon monoxide (CO), 0.4% nitric oxide (NO), and 80% oxygen (HI-OX) packages over 120 h (*p* < 0.001; SEM = 0.415). ^1^ 1 = very bright red, 2 = bright red, 3 = dull red, 4 = slightly dark red, 5 = moderately dark red, 6 = dark red to tannish-red, 7 = dark reddish-tan, and 8 = tan to brown; (**b**) Least squares means of percent discoloration for CO, NO, and HI-OX packages over 120 h (*p* < 0.001; SEM = 5.436).

**Figure 2 foods-13-00902-f002:**
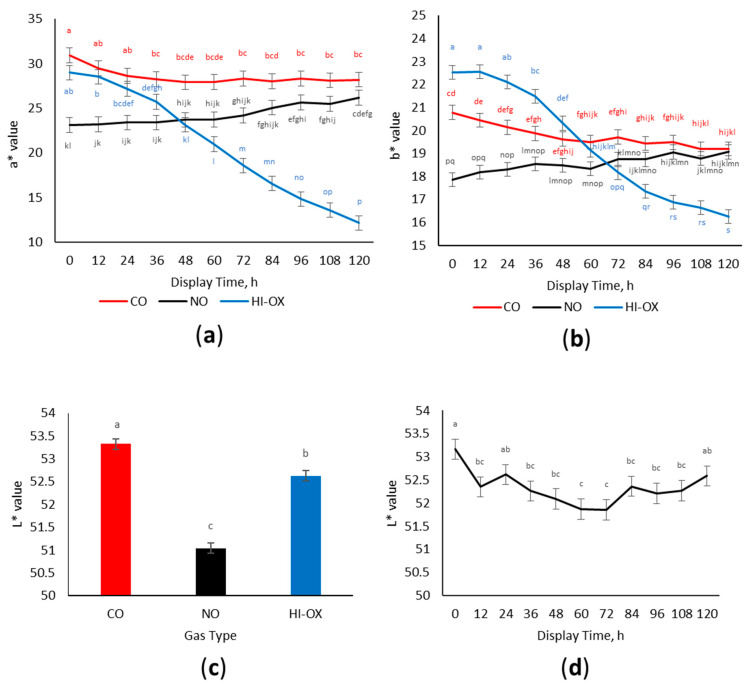
Instrumental color data. LS means with the same superscript are not significantly different (*p* > 0.05): (**a**) Least squares means of a* for 0.4% carbon monoxide (CO), 0.4% nitric oxide (NO), and 80% oxygen (HI-OX) packages over 120 h (*p* < 0.001; SEM = 0.829); (**b**) Least squares means of b* for CO, NO, and HI-OX packages over 120 h (*p* < 0.001; SEM = 0.302); (**c**) Least squares means of L* for CO, NO, and HI-OX packages (*p* < 0.001; SEM = 0.113); (**d**) Least squares means of L* over 120 h for all packaging treatments (*p* = 0.002; SEM = 0.216).

**Figure 3 foods-13-00902-f003:**
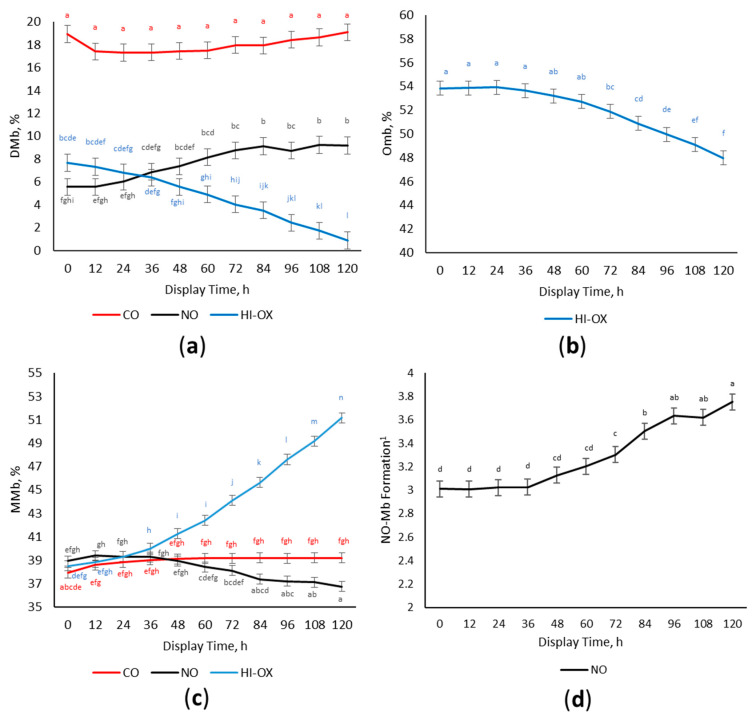
Percentage of surface myoglobin content. LS means with the same superscript are not significantly different (*p* > 0.05): (**a**) Least squares means of deoxymyoglobin (DMb) for 0.4% carbon monoxide (CO), 0.4% nitric oxide (NO), and 80% oxygen (HI-OX) packages over 120 h (*p* < 0.001; SEM = 0.734); (**b**) Least squares means of oxymyoglobin (OMb) for HI-OX packages over 120 h (*p* < 0.001; SEM = 0.586); (**c**) Least squares means of metmyoglobin (MMb) for CO, NO, and HI-OX packages over 120 h (*p* < 0.001; SEM = 0.432); (**d**) Least squares means of NO-Mb formation for NO packages over 120 h (*p* < 0.001; SEM = 0.068). ^1^ NO-Mb formation is expressed as the ratio of absorbance at wavelengths of 650 nm over 570 nm. A greater number indicates more NO-Mb formation.

**Figure 4 foods-13-00902-f004:**
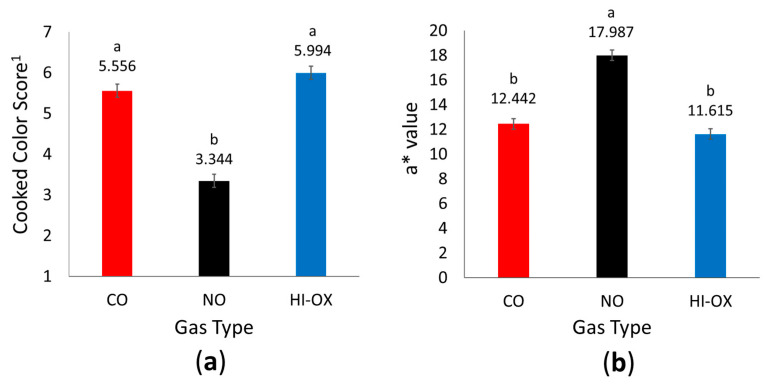
Internal cooked color data. LS means with the same superscript are not significantly different (*p* > 0.05): (**a**) Least squares means for internal cooked color score for 0.4% carbon monoxide (CO), 0.4% nitric oxide (NO), and 80% oxygen (HI-OX) packages (*p* < 0.001; SEM = 0.161). ^1^ 1 = very red, 2 = slightly red, 3 = pink, 4 = slightly pink, 5 = pinkish-gray, 6 = grayish tan/brown, 7 = tan/brown; (**b**) Least squares means for a* values of cooked internal cross-sections for CO, NO, and HI-OX packages (*p* < 0.001; SEM = 0.425).

**Figure 5 foods-13-00902-f005:**
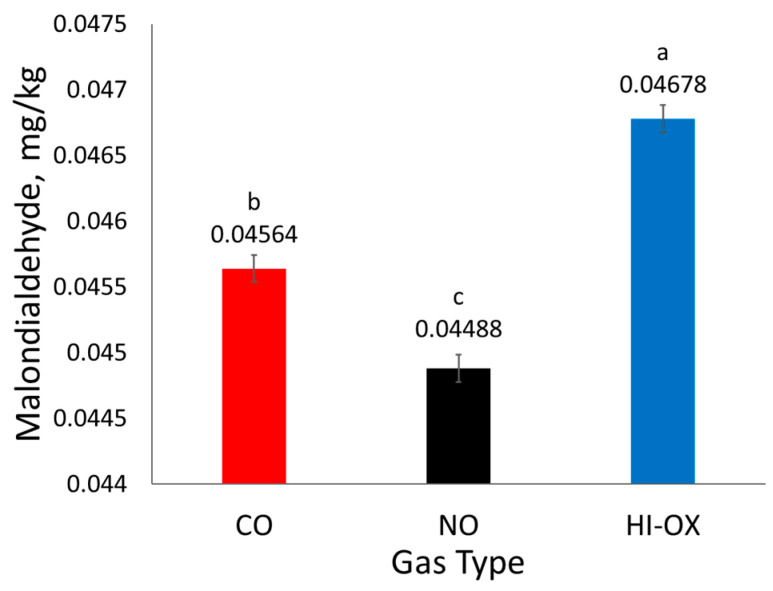
Least squares means of mg malondialdehyde/kg meat homogenate for 0.4% carbon monoxide (CO), 0.4% nitric oxide (NO), and 80% oxygen (HI-OX) packages (*p* < 0.001; SEM = 0.000104). LS means with the same superscript are not significantly different (*p* > 0.05).

**Figure 6 foods-13-00902-f006:**
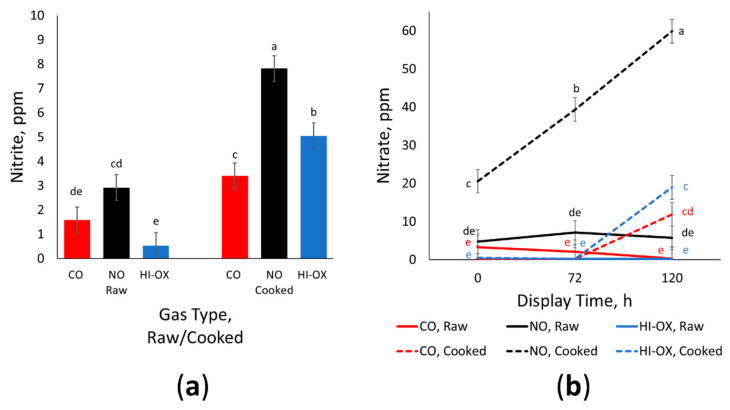
Residual nitrite and nitrate data. LS means with the same superscript are not significantly different (*p* > 0.05): (**a**) Least squares means of ppm residual nitrite for raw and cooked 0.4% carbon monoxide (CO), 0.4% nitric oxide (NO), and 80% oxygen (HI-OX) samples (*p* = 0.009; SEM = 0.531); (**b**) Least squares means of ppm residual nitrate for raw and cooked CO, NO, and HI-OX samples at 0, 72, and 120 h (*p* < 0.001; SEM = 3.098).

**Table 1 foods-13-00902-t001:** AMSA 8-point scale for ground beef display discoloration.

Score	Color Description
1	= very bright red
2	= bright red
3	= dull red
4	= slightly dark red
5	= moderately dark red
6	= dark red to tannish-red
7	= dark reddish-tan
8	= tan to brown

**Table 2 foods-13-00902-t002:** AMSA 7-point scale for internal cooked color.

Score	Color Description
1	= very red
2	= slightly red
3	= pink
4	= slightly pink
5	= pinkish-gray
6	= grayish tan/brown
7	= tan/brown

## Data Availability

The original contributions presented in the study are included in the article, further inquiries can be directed to the corresponding author.

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
