# Peer review of "Comparison of Gas Treatments of High Oxygen, Carbon Monoxide, and Nitric Oxide on Ground Beef Color in Modified Atmosphere Packaging"

_foods, 2024, doi:10.3390/foods13060902_

Round 1

Reviewer 1 Report

Comments and Suggestions for Authors

The manuscript titled "Comparison of gas treatments of high oxygen, carbon monoxide, and nitric oxide on ground beef color in modified atmosphere packaging" submitted by Carpenter et al., has been revised. The paper compares the quality of ground meat stored in different modified atmosphere packaging (MAP) conditions. The study is interesting because it provides insights into using NO-MAP as an alternative to HI-OX and CO-MAP for ground beef. The overall presentation of the paper is well-written and well-organized, and the experiment is well-designed. However, this reviewer has a few suggestions to improve the manuscript.

Firstly, this reviewer recommends including details about the ground meat's characteristics, such as the muscle type, time postmortem, pH, and fat content, to better understand the study (Line 80). Additionally, it is suggested that a few minor edits be made, such as changing "minutes" to "min," (Lines 146, 148, 163), "seconds" to "s," (Line 162), and separating the temperature unit (°C) from the number (4 °C) (Lines 160, 162). Replacing "hour" with "h" in lines (L203, L205, L224-226, L253-255) is also recommended. Finally, it is recommended that a conclusion title be included in the last paragraph.

Author Response

The manuscript titled "Comparison of gas treatments of high oxygen, carbon monoxide, and nitric oxide on ground beef color in modified atmosphere packaging" submitted by Carpenter et al., has been revised. The paper compares the quality of ground meat stored in different modified atmosphere packaging (MAP) conditions. The study is interesting because it provides insights into using NO-MAP as an alternative to HI-OX and CO-MAP for ground beef. The overall presentation of the paper is well-written and well-organized, and the experiment is well-designed. However, this reviewer has a few suggestions to improve the manuscript.

 Firstly, this reviewer recommends including details about the ground meat's characteristics, such as the muscle type, time postmortem, pH, and fat content, to better understand the study (Line 80).

Ground beef was specified as “commodity 80/20 ground beef.” Ground beef chubs were purchased from a commercial purveyor where beef trimming was used for the blend, therefore muscle type and time postmortem are unknown and not stated. Fat content is now specified as 20%, but the pH was not measured but should be assumed similar between chubs. If there was any variation in fat content or pH between the purchased chubs, it would be accounted for by the block design of the statistical analyses.

Additionally, it is suggested that a few minor edits be made, such as changing "minutes" to "min," (Lines 146, 148, 163), "seconds" to "s," (Line 162), and separating the temperature unit (°C) from the number (4 °C) (Lines 160, 162). Replacing "hour" with "h" in lines (L203, L205, L224-226, L253-255) is also recommended.

Suggested edits were made where applicable.

Finally, it is recommended that a conclusion title be included in the last paragraph.

The last paragraph of the discussion was separated under a “conclusions” heading.

Reviewer 2 Report

Comments and Suggestions for Authors

1、Please provide the most important data with refined statement rather than a talk in generalities in abstract. The content includes purpose, method, result, conclusion in abstract.

2、The introduction needs to be enriched. Why did you choose nitric oxide MAP to package for ground beef?

3、Whether the method of 2.2 and 2.4 is objective?

4、What is the final conclusion of this study? Whether nitric oxide MAP can be used as the alternative to high oxygen and carbon monoxide MAP for ground beef. Please add the conclusion of the article.

5、The description of Line 403-409 “Although CO would still be preferable over NO  to maintain redness for MAP ground beef, NO exhibits a unique color cycle warranting further research to optimize its concentration and use. Along with an antioxidant effect, NO could have a possible antimicrobial effect which should be investigated. The pathway of the reduction of NO-MMb to NO-Mb also needs further investigation to better understand the mechanism for the color cycle of nitric oxide and myoglobin” need more references. This part does not present a clear result of the paper. It's a very vague description.

Comments on the Quality of English Language

The English language needs to improve throughout the manuscript.

Author Response

1、Please provide the most important data with refined statement rather than a talk in generalities in abstract. The content includes purpose, method, result, conclusion in abstract.

            More definitive data statements for analyses interactions and main effects were added to the abstract with significance specified with stated P-values. In order to stay within the 200 word limit for MDPI abstracts, the first 2 sentences of the original abstract introducing the study were removed.

2、The introduction needs to be enriched. Why did you choose nitric oxide MAP to package for ground beef?

       Along with the tilapia study from Wang et al., the ability of nitric oxide to bind with myoglobin is discussed and provided as reasoning for why NO-MAP was chosen for ground beef.

3、Whether the method of 2.2 and 2.4 is objective?

       The topic of if trained color panels are objective evaluations of color can be debated, but according to the 2023 AMSA color guidelines, it is objective when panelists are properly trained. Because of this, panelists were specified as being trained for “objective evaluation” in method 2.2.

4、What is the final conclusion of this study? Whether nitric oxide MAP can be used as the alternative to high oxygen and carbon monoxide MAP for ground beef. Please add the conclusion of the article.

            The last paragraph of the discussion was separated into its own conclusion section. It was then further expanded upon and split into two paragraphs. The first paragraph states the results of this study and directly states the conclusion that CO outperformed NO and HI-OX for this display setting. It also states that NO is less desirable due to its persistent pinking and increased residual nitrates and nitrites.

            The second paragraph of the conclusion accounts is the most similar to what was previously in the draft. It states that NO did not outperform CO and HI-OX but still has viability as a MAP gas type. It states how future usage of NO could be altered to possibly optimize its use, and what research should be done to better understand its effects on quality and shelf life, as well as its mechanism in the myoglobin color cycle.

5、The description of Line 403-409 “Although CO would still be preferable over NO  to maintain redness for MAP ground beef, NO exhibits a unique color cycle warranting further research to optimize its concentration and use. Along with an antioxidant effect, NO could have a possible antimicrobial effect which should be investigated. The pathway of the reduction of NO-MMb to NO-Mb also needs further investigation to better understand the mechanism for the color cycle of nitric oxide and myoglobin” need more references. This part does not present a clear result of the paper. It's a very vague description.

            The addition of the first paragraph of the conclusion (lines 403-413) directly states the results of the paper and our objective.

            Nitric oxide’s antioxidant effect is supported by the decreased TBARS values of this study, but the possible antimicrobial effect is supported through reference to one of the paper’s citations.

Comments on the Quality of English Language

The English language needs to improve throughout the manuscript.

            The paper was revised and edits were made where appropriate.

Round 2

Reviewer 2 Report

Comments and Suggestions for Authors

Paper has been revised and can be accepted.